

# Dominance relationships in a family pack of captive arctic wolves (*Canis lupus arctos*): the influence of competition for food, age and sex

Simona Cafazzo[1,2], Martina Lazzaroni[1,2] and Sarah Marshall-Pescini[1,2]

[1] Wolf Science Centre, Ernstbrunn, Austria

[2] Comparative Cognition, Messerli Research Institute, University of Veterinary Medicine Vienna, Medical University of Vienna, University of Vienna, Vienna, Austria

## ABSTRACT

**Background**. Dominance is one of the most pervasive concepts in the study of wolf social behaviour but recently its validity has been questioned. For some authors, the bonds between members of wolf families are better described as parent-offspring relationships and the concept of dominance should be used just to evaluate the social dynamics of non-familial captive pack members (e.g., *Mech & Cluff, 2010*). However, there is a dearth of studies investigating dominance relationships and its correlates in wolf family packs.

**Methods**. Here, we applied a combination of the most commonly used quantitative methods to evaluate the dominance relationships in a captive family pack of 19 Arctic wolves.

**Results**. We found a significant linear and completely transitive hierarchy based on the direction of submissive behaviours and found that dominance relationships were not influenced by the competitive contexts (feeding vs. non-feeding context). A significant linear hierarchy also emerges amongst siblings once the breeding pair (the two top-ranking individuals) is removed from analyses. Furthermore, results suggest that wolves may use greeting behaviour as a formal signal of subordination. Whereas older wolves were mostly dominant over younger ones, no clear effect of sex was found. However, frequency of agonistic (submissive, dominant and aggressive) behaviours was higher between female–female and male–male dyads than female–male dyads and sex-separated linear hierarchies showed a stronger linearity than the mixed one. Furthermore, dominance status was conveyed through different behavioural categories during intra-sexual and inter-sexual interactions.

**Discussion**. Current results highlight the importance of applying a systematic methodology considering the individuals' age and sex when evaluating the hierarchical structure of a social group. Moreover, they confirm the validity of the concept of dominance relationships in describing the social bonds within a family pack of captive wolves.

Corresponding author
Simona Cafazzo,
simona.cafazzo@gmail.com

## INTRODUCTION

In group-living animals, the natural asymmetries among individuals in their ability to prevail in competition may result in social dominance hierarchies. In general, dominant-subordinate relationships may be defined as long-term dyadic relationships characterised by an asymmetric distribution of agonistic behaviours (*Drews, 1993*). In line with this definition, dominance relationships are measured in terms of the degree of the unidirectionality of behaviours exhibited within a dyad (*directional consistency index, Van Hooff & Wensing, 1987*), a high unidirectionality for example, would emerge if A frequently showed submissive behaviours to B, but B was never observed showing a submissive behaviour to A. When dominant-subordinate relationships characterize all or most dyads in a social group, then it may be possible to describe the overall social structure of that group as a '*linear dominance hierarchy.*' To fit the linear hierarchy model, all or most relationships between group members have to be transitive (A > B > C > D), rather than circular (A > B and B > C but C > A), (*Landau, 1951*; *Appleby, 1983*; *Van Hooff & Wensing, 1987*; *De Vries, 1995*; *Shizuka & McDonald, 2012*; *Shizuka & McDonald, 2014*). However, a major limitation of the linearity index is that it becomes negatively biased when some pairs of individuals fail to interact (null dyads; *De Vries, 1995*; *Klass & Cords, 2011*). Therefore, a low level of linearity may not necessarily indicate the absence of transitivity but may be due to the high percentage of null dyads. More recently, *triangle transitivity* ($t_{tri}$), which determines the level of hierarchy transitivity and is less sensitive to unknown relationship (*Shizuka & McDonald, 2012*; *Shizuka & McDonald, 2014*), has been used as a successful alternative (e.g., *Norscia & Palagi, 2015*). Triangle transitivity is based on the transitivity of dominance relations among sets of three individuals that all interact with each other (*Shizuka & McDonald, 2012*). The method by *Shizuka & McDonald (2012)* follows logic similar to that of *De Vries (1995)*, but the procedure is conducted without filling in zero dyads (i.e., unknown relationships) with randomized dominance relations. In fact, filling in zero dyads artificially decreases the level of linearity because it creates cyclic (and not transitive) triads, e.g., A dominates B, B dominates C, and C dominates A (A > B > C > A). Triangle transitivity and linearity are equivalent when the dominance relations of all dyads are known but since such complete observations are rare in empirical studies (*Shizuka & McDonald, 2012*); using both measures may provide a better assessment of the hierarchical organization of a group.

As clearly demonstrated by *Shizuka & McDonald (2015)*, another factor affecting the linearity is the interaction rates of the top-ranked individuals. In fact, in groups where alpha individuals engaged in more contests, there were more double-dominant triads (where A dominates both B and C and any relationships is detectable between B and C) and fewer pass-along triads (where A dominates B, B dominates C but any relationships is detectable between A and C), suggesting that top-ranking individuals may have a disproportionate influence on dominance hierarchies and highlighting the necessity to control for that (*Shi, Hanser & McHugh, 2009*; *Modlmeier et al., 2014*; *Shizuka & McDonald, 2015*).

When a significant degree of linearity/transitivity exists in a group, the rank order most consistent with a linear hierarchy may be obtained via *the binary dyadic dominance relationship method* (I&SI, *De Vries, 1998*), which allows the characterization of the relationship between two individuals in a group. However, the dyadic relationship between individuals may very across contexts since the capacity and motivation of animals to obtain access to specific resources (e.g., feeding, mating privileges etc.) may vary considerably. Since the costs and benefits of winning a given conflict may be context-dependent, a dyad member may be dominant in one context but not in others (*Hand, 1986*), thereby influencing the overall agonistic rank position of members in a pack. Such differences in rank relationships depending on context have indeed been reported for several species ranging from chimpanzees to cats (*Noë, De Waal & Van Hooff, 1980*; *Bonanni et al., 2007*).

A hierarchy within a group can be described in terms of the asymmetry amongst individuals in winning conflicts (agonistic dominance), however it can also be described in terms of *formal dominance* (*De Waal, 1989*). The latter is characterized by the exchange of ritualized signals and/or greeting rituals, the direction of which is independent of the social context. When the agonistic dominance is accepted by the subordinates, dominance relationships are stable (i.e., no reversals or intransitivity in the hierarchy emerge; e.g., *Fournier & Festa-Bianchet, 1995*), and formal and agonistic dominance coincide. In this case, the exchange of hierarchical status information may be conveyed mainly through *formalized submissive signals*, resulting in a low frequency of overt aggressive conflicts (*De Waal, 1989*).

Wolves are a particularly interesting case in which to investigate hierarchical relationships since the applicability of this concept to this species has been suggested to depend largely on the type of social structure of the pack (e.g., *Bloch, 2002*; *Mech & Cluff, 2010*). The *typical wild wolf social structure* is based around a bonded male–female pair that raises pups communally. The offspring of a bonded pair may forego dispersal and remain with their native pack and help raise later litters (e.g., *Mech, 1999*; *Mech, 2000*; *Packard, 2003*). Traditionally, dominance relationships in wild wolf packs have been described in terms of the "*age(sex)-graded model*[1]" (*Zimen, 1982*; *Packard, 2003*, pp 54) in that within these family groups, there is a natural dominance order in which offspring submit to parents, and puppies submit to both parents and older siblings (*Mech, 1999*). This model has been presented in one of two ways: either simply as separate linear hierarchies for each sex influenced but not determined by age (*Schenkel, 1967*; *Zimen, 1982*) or as male dominance over females within each age class (*Rabb, Woolpy & Ginsburg, 1967*; *Fox, 1980*; *Zimen, 1982*; *Van Hooff & Wensing, 1987*; *Savage, 1988*).

In contrast to this traditional "age-(and sex)-graded" model, more recently several authors claim that the relationships in the typical wolf family, are better described as parents-offspring-pups relationships rather than as a pecking order dominated by an alpha male (alpha, beta down to omega animals; e.g., *Bloch, 2002*; *Fatjo et al., 2007*; *Mech & Cluff, 2010*). Thus, a dominance hierarchy within the *typical 'nuclear' family pack* could be alternatively viewed as just a reflection of the age, sex and reproductive structure of the group (*Mech, 1999*). Although the same authors suggest that traditional concepts of

[1] The model was originally called the "age-graded model" but since it entails also an effect on sex on dominance relationships for a major clarity we prefer refers to it as the "age-(sex)-graded model."

dominance hierarchy may still be useful for evaluating the social dynamics of captive families composed of artificially assembled unrelated individuals (*Mech, 1999*; *Packard, 2003*; *Fatjo et al., 2007*).

To date, although dominance relationships have been described in several packs of wild wolves (e.g., *Mech, 1999*; *Bloch, 2002*; *Peterson et al., 2002*; *Sands & Creel, 2004*; *Baan et al., 2014*), a quantitative assessment and statistical validation of the existence of linear dominance relationships has been carried out only in wolves living in captivity and mostly in packs consisting of either 'complex families' i.e., composed of artificially assembled unrelated individuals (e.g., *Packard, 2003* for a review; *Mazzini et al., 2013*) or 'disrupted families' i.e., in which one or both of the original parents are missing (e.g., *Packard, 2003* for a review; *Cordoni & Palagi, 2008*). To our knowledge, only two studies systematically tested and confirmed the existence of a linear hierarchy between pack members in typical wolf families ('nuclear families' with parents and multiple generations of offspring and 'extended families' consisting of parents plus one of more of their siblings, and their direct offspring) (*Van Hooff & Wensing, 1987*; *Romero et al., 2014*).

Formal signals of dominance or submission have been described in wild wolves and captive wolves (*Schenkel, 1947*; *Feddersen-Petersen, 2004*), for example *Van Hooff & Wensing (1987)*, found that postural displays (namely high posture and low posture) were exhibited consistently from one partner to the other and were therefore suggested as indicators of formal dominance. However, signals of formal dominance have so far been systematically tested only in domestic dogs (*Cafazzo et al., 2010*; *Van der Borg et al., 2015*). In particular, mouth licking associated with tail wagging (so called 'greeting' behaviour) occurring often during greeting ceremonies fulfilled the criteria of a formal signal of subordination in both free-ranging (*Cafazzo et al., 2010*) and group housed dogs (*Van der Borg et al., 2015*).

In previous studies, although a linear hierarchy clearly emerged within the pack, there was no quantitative assessment of the validity of the ''age-(sex)-graded' model. Indeed, although female and male separate linear hierarchies have been commonly described qualitatively in captive wolves (e.g., *Schenkel, 1947*; *Zimen, 1982*) results tend to go in the opposite direction to descriptions of social relationships between the sexes in wild wolves (e.g., *Clark, 1971*; *Haber, 1977*; *Mech, 1999*). However, no study has been carried out to assess the validity of this model neither in the wild nor in captivity. Furthermore, despite having been observed in a number of species, no study, to our knowledge has evaluated the potential effect of different competitive contexts on dominance relationships in wolves.

Hence, in the current study we applied a combination of the most commonly used quantitative methods to evaluate dominance relationships, in order to investigate (1) whether an agonistic dominance hierarchy could describe the relationships between members of a captive family pack of Arctic wolves and whether a hierarchical structure would remain consistent (amongst siblings) when the top-ranking individuals (i.e., the breeding pair) were removed from analyses (see *Shizuka & McDonald, 2015*); (2) whether the hierarchy remained consistent in a feeding and non-feeding context; (3) which behaviours may be the best indicators of dominance (4) whether mouth licking associated

to tail wagging (greeting, hereafter) can be considered a formal signal of submission in wolves, as has been found in dogs. Finally, to help address the controversy regarding the validity of the dominance concept in wolves, we also aimed to assess (5) if the typical age- (and sex)-graded model could adequately describe the relationships of the nuclear family pack of wolves observed.

## MATERIALS & METHODS

### Ethics statement

The study was purely observational with no manipulation of animals. The relevant committee, Tierversuchs-kommission am Bundesministerium für Wissenschaft und Forschung (Austria) allows us running this research without special permissions regarding animals (wolves) since this is not required in such socio-cognitive studies in Austria (Tierversuchsgesetz 2012–TVG 2012).

### Subjects and study site

We studied a pack of captive arctic wolves (*Canis lupus arctos*) at the Olomouc Zoo (Moravia, Czech Republic) during a period of five months (Jan 2014–May 2014). The pack was composed of 20 individuals: seven adult males, eight adult females, three sub-adult males, and eight sub-adult females. Adults were defined as individuals older than two years and sub-adults as individuals younger than two years. The pack was structured as a nuclear family with all members born into the pack except for the breeding male and the two unrelated breeding females. The number of individuals decreased to 14 by the end of the period of study (Table 1) because six wolves were removed from the pack. Specifically, one adult breeding female was removed because she was badly injured by the other breeding female during the reproductive season. Two adult and two sub-adult wolves were sold to another zoo. Finally an adult male was removed because of continuous mobbing episodes from the whole pack.

The pack was kept from February to March in an enclosure of approximately 7,000 m$^2$ located in a naturally hilly area equipped with trees, branches and dens. For the rest of the study period the animals were restricted in a smaller enclosure of about 3,000 m$^2$. The animals were fed with pieces of meat, which were put on a table of 2 m$^2$, four or five times every week in the early afternoon. Water was available ad libitum. No stereotypic or aberrant behaviours characterized the study group.

### Data collection

The pack was observed six days per week for 2–3 h a day, either in the morning or the afternoon (the afternoon period included feeding time). Before commencing systematic data collection, the observer (M.L.) underwent a 7 month training period on wolf behaviour whilst collecting data at the Wolf Science Centre (Ernstrbunn, Austria). Then she carried out some preliminary observations (50 h) in order to (1) identify all individuals belonging to the pack and (2) establish the data collection methods.

Wolf behaviour was observed in two different social contexts: in the presence of food and in the absence of this source of competition. Data collection was carried out following

Table 1 **The group of arctic wolves (*C. lupus arctos*) housed at the Olomouc Zoo (Moravia, Czech Republic).** The animals were classified adult when older than two years and sub-adult when younger than two years.

| Name | Sex | Date of birth | Age class | Removed | Returned |
|---|---|---|---|---|---|
| Viki | Female | Mar-04 | Adult | | |
| Beta[a] | Female | Apr-07 | Adult | 09-Feb | |
| Normale | Female | Apr-12 | Sub-adult | 13-Mar | |
| Uno Beta | Female | Apr-12 | Sub-adult | | |
| Lacrima | Female | Apr-12 | Sub-adult | | |
| Volpe | Female | Apr-12 | Sub-adult | 22-Apr | 12-May |
| Musocorto | Female | Apr-12 | Sub-adult | | |
| Husky | Female | Apr-13 | Sub-adult | 13-Mar | |
| Sosia | Female | Apr-13 | Sub-adult | | |
| Cane | Female | Apr-12 | Sub-adult | | |
| Macchia | Male | Mar-04 | Adult | | |
| Maschera | Male | Apr-09 | Adult | 13-Mar | |
| Sfregiato | Male | Apr-09 | Adult | | |
| Storto | Male | Apr-09 | Adult | 13-Mar | |
| Procione | Male | Apr-10 | Adult | 22-Apr | |
| Secondo | Male | May-11 | Adult | | |
| Taglio | Male | Apr-12 | Sub-adult | | |
| Musolungo | Male | May-11 | Adult | | |
| Due | Male | Apr-12 | Sub-adult | | |
| Zampa | Male | Apr-12 | Sub-adult | | |

**Notes.**

[a] This female was excluded from the analyses since she was removed after a short time from the beginning of the study and her behavioural data were insufficient.

*Altmann*'s *(1974)* methods: the focal animal sampling method was used in the absence of sources of competition, whereas the subgroup animal sampling method was used in the presence of food. During focal-subgroup sessions all occurrences of agonistic and greeting interactions were recorded (Table 2). The "*ad libitum*" sampling method (*Altmann, 1974*) was also used to record all agonistic behavioural patterns as well as greetings occurring out of focal-subgroup sampling sessions; we gathered 154 h of observations, distributed over 63 days.

Each individual measure of all behaviour patterns was corrected for animal observation time because the latter varied between individuals; since some animals were removed from the pack earlier.

## Agonistic dominance hierarchy and behavioural analyses

In order to determine the agonistic dominance hierarchy, the outcomes of aggressive, submissive, and dominance dyadic interactions (see Table 2 for a description) were ranked in three different squared matrices with winners on one axis and losers on the other. This procedure was applied to interactions in both social contexts (i.e., presence and absence of food), resulting in a total of six matrices, three for each context. In the same way, in order to analyse the use of greeting as a formal signal of subordination,

**Table 2** Ethogram.

| Behavioural categories | Behavioural pattern | Description |
|---|---|---|
| **Greeting behaviours** | Greeting | To interact in a friendly and relaxed manner holding the ears back, showing much tail wagging and licking of the other's mouth/muzzle. The subject however does not show crouching/lowered hindquarters nor is the tail tucked between the legs |
| **Dominance behaviours** | Stand tall | Subject straightens up to full height, with a rigid posture and tail, may include raised hackles, ears erect and tail perpendicular or above the back |
| | Stand over | To stand over another's body, with all four paws on the ground and the tail above the plane of the back. The receiver may have either the whole body or just the forepaws under the actors' belly/side |
| | Paw on | To place one or both forepaws on the other's back |
| | Ride up | To mount another one from behind or from the side, exhibiting a thrusting motion |
| | Head on | The subject approaches another's shoulder/back with the tail above the plane of the back and puts its head on it. Most of times formation looks like a capital "T" |
| | Muzzle bite | To grab the muzzle of another subject softly |
| | Approach dominant | To approach another subject within one body length for at least 5 s, with the tail perpendicular or above the plane of the back and the ears erect and pointed forward |
| **Submissive behaviours** | Crouch | Lowering the head, sometimes bending the legs, arching the back, lowering the tail between the hind legs, and avoiding eye contact |
| | Passive submission | To lie on the back showing the stomach and holding the tail between the legs. The ears are held back and close to the head and the subject raises a hind leg for inguinal presentation |
| | Active submission | The subject has its tail tucked between the hind legs sometimes wagging it while he is in a crouched position (with hindquarters lowered) and may attempt to paw and lick the side of actors'/aggressor's muzzle. The behaviour may include urination |
| | Withdrawing | The subject withdraws from another moving away slowly in the opposite direction, displaying a submissive posture. It occurs when a subject has been threatened or attacked by another, or a fight has taken place |
| | Flee | To run away from another with tail tucked between the legs and body ducked. It occurs when a subject has been threatened or attacked by another, or after a fight |
| | Avoidance | In response to another reducing the distance towards it, the subject moves away displaying a submissive posture. The subject may also look at the individual he is trying to avoid |
| | Approach submissive | To slowly approach another within one body length remaining within that distance for at least 5 s. The approach is characterized by a ducked posture and tail between the legs. Subject can also be moving in a wavy line and in a hesitant (stop-start) manner |
| **Aggressive behaviours** | Threat | Subject orients towards another performing one or more of the following: staring intently at, curling of the lips, baring of the canines, raising the hackles, snarling, growling, and barking, sometimes with the tail perpendicular or above the back |
| | Attack | Running into or jumping onto another with tail, ears and sometimes hackles up, often with bites at the neck |
| | Knock down | To strike another subject sharply with the chest or shoulder so that the other falls to the ground |
| | Pin | To grab another at the neck or at the muzzle, forcing it down to the ground and holding it there |
| | Chase | A subject runs after a conspecific, exhibiting threatening behaviours (see 'threat' above) |
| | Snapping | To snap teeth into the air, noisily |
| | Bite | Bite a conspecific, without inhibition, with enough pressure to cause potential injury |

all greeting interactions observed (Table 2) were ranked in a squared matrix. For each matrix, linearity and transitivity (and their statistical significance), as well as directional consistency, were calculated.

The values of the Landau's corrected linearity index h′, Triangle transitivity ($t_{tri}$), and DCI ranged from 0 to 1. Values of 0 indicate a complete absence of linearity and transitivity (i.e., no hierarchy) and a complete bidirectionality; values of 1 indicate a perfect linear hierarchy, the absence of circular triads and a complete unidirectionality.

When significant linearity was detected, dominance ranks were determined using the I&SI method which minimizes inconsistencies and strengths of inconsistencies in dominance relationships (*De Vries, 1998*). An inconsistency occurs when individual j dominates i, and j's rank is lower than i's (*De Vries, 1998*). The rank difference between two individuals involved in an inconsistency is the strength of that inconsistency (*De Vries, 1998*).

The dominance rank order obtained for each matrix was standardized by distributing ranks evenly between the highest (+1) and the lowest (−1), with the median rank being scored as 0 (*East & Hofer, 2001*). Then, in order to test the effect of social context, age and gender, we calculated the Spearman's correlation coefficient between (1) I&SI rank orders in the two competitive contexts, (2) between inter-sexual and intra-sexual hierarchies, and (3) between rank orders and age.

To classify greeting as a formal indicator of subordination we assessed whether it fit specific criteria: (a) completely unidirectional (DCI = 1) (b) shown between most of the pack members (no null dyads) and (c) correlation with dominance relationships based on agonistic behaviours (*Waal & Luttrell, 1985*; *Waal, 1986*; *De Waal, 1989*; *Preuschoft, 1999*; *Vervaecke, De Vries & Van Elsacker, 2000*).

All correlation analyses were plotted and visually checked to ascertain that data were not clustered

Linearity, DCI and I&SI rank orders were calculated using Matman 1.1 (10.000 randomizations; Noldus Information Technology, Wageningen, The Netherlands; (*Vries, Netto & Hanegraaf, 1993*). Spearman's rank correlations were calculated in STATISTICA 7.1 edition (*StatSoft Italia srl, 2005*). We calculated the proportion of transitive triangles relative to all triangles (Pt), the triangle transitivity metric ($t_{tri}$) and its statistical significance using the codes provided in *Shizuka & McDonald (2012)*; supplementary material; errata corrige: (*Shizuka & McDonald, 2014*; package 'statnet' *Handcock et al., 2015*) in the R programme version 3.2.3 (*R Development Core Team, 2011*). The alpha male and female totalized just under half of our data points in all three categories of behaviours considered (41.84% of dominance behaviours; 44.06% of aggressive behaviors; 51.46% of submissive behaviors received). Hence, taking into account the potential disproportionate role of these individuals in affecting the linearity of the hierarchy (as suggested by *Shizuka & McDonald, 2015*) we re-ran the analyses described above without these two animals.

In order to investigate whether the dyadic distribution of greeting was influenced by rank, age and sex we ran a generalized linear mixed model (GLMM with a Poisson distribution) with the frequency of greeting behaviour as the response factor and sex

**Table 3 Different dominance measures.** Summary of values of the directionality (directional consistency index, DCI), linearity h′ and its significance level, number and strength of inconsistencies (No. I and SI, respectively) for the I&SI rank orders, and triangle transitivity (proportion of transitive triangles relative to all triangles $P_t$, triangle transitivity metric $t_{tri}$, and significance level) for all behavioural categories in the absence, and in the presence, of food and for all dominance and submissive interactions regardless of the context.

| | Directionality | Linearity | Inconsistency: No. I (and SI) | Triangle transitivity |
|---|---|---|---|---|
| **Agonistic behaviours displayed in the absence of food** | | | | |
| Aggressive behaviour | DCI = 0.94 | No (h′= 0.25, p = 0.08) | – | $P_t = 0.88$, $t_{tri} = 0.52$, $p = 0.009$ |
| Dominance behaviour | DCI = 0.97 | Yes (h′ = 0.38, p = 0.002) | 1 (3) | $P_t = 0.99$, $t_{tri} = 0.97$, $p = 0.000$ |
| Submissive behaviour | DCI = 0.98 | Yes (h′ = 0.45, p = 0.0001) | 1 (2) | $P_t = 0.99$, $t_{tri} = 0.97$, $p = 0.000$ |
| **Agonistic behaviours displayed in the presence of food** | | | | |
| Aggressive behaviour | DCI = 0.85 | Yes (h′ = 0.36, p = 0.003) | 4 (24) | $P_t = 0.98$, $t_{tri} = 0.90$, $p = 0.000$ |
| Dominance behaviour | DCI = 0.99 | Yes (h′ = 0.33, p = 0.006) | 1 (4) | $P_t = 1.00$, $t_{tri} = 1.000$, $p = 0.000$ |
| Submissive behaviour | DCI = 0.99 | Yes (h′ = 0.30, p = 0.02) | 1 (2) | $P_t = 0.99$, $t_{tri} = 0.95$, $p = 0.000$ |
| **Agonistic behaviours displayed in both the absence and the presence of food** | | | | |
| All dominance behaviours | DCI = 0.97 | Yes (h′ = 0.58, p = 0.0001) | 2 (9) | $P_t = 0.992$, $t_{tri} = 0.970$, $p = 0.000$ |
| All submissive behaviours | DCI = 0.97 | Yes (h′ = 0.56, p = 0.0001) | 0 | $P_t = 1.000$, $t_{tri} = 1.000$, $p = 0.000$ |
| **Agonistic behaviours displayed in both the absence and the presence of food without considering the alpha male and female** | | | | |
| All dominance behaviours | DCI = 0.96 | Yes (h′ = 0.44, p = 0.0008) | 2 (9) | $P_t = 0.983$, $t_{tri} = 0.932$, $p = 0.000$ |
| All submissive behaviours | DCI = 0.94 | Yes (h′ = 0.40, p = 0.005) | 0 | $P_t = 1.000$, $t_{tri} = 1.000$, $p = 0.000$ |

combination of the dyad (female–female, male–male and female–male), age difference expressed in months and rank relationship between actor and receiver as independent factors. The identity of the actor and the dyad were entered as random factors. Finally, to investigate the effect of sex on the distribution of agonistic behaviours we ran a generalized linear mixed model (GLMM with a Poisson distribution) with the frequency of agonistic behaviour as the response factor and gender combination of the dyad (female–female, male–male and female–male) as the independent factor The identity of both individuals in the dyad was entered as random factors.

The generalized linear mixed models were calculated using the codes provided in lme4-package (*Bates et al., 2015*), in R version 3.2.3 (*R Development Core Team, 2011*).

## RESULTS

### Agonistic dominance relationships in the absence and presence of food

The directional consistency (DCI), linearity (h′), the number and strength of inconsistency (I&SI), the proportion of transitive triangles relative to all triangles (Pt), and the triangle transitivity metric ($t_{tri}$) for each agonistic behavioural category in the two different contexts analysed (in the presence of food and in its absence) are summarised in Table 3.

#### Agonistic dominance in the absence of food

Aggressive interactions ($N = 388$) did not show significant linearity; nevertheless the behaviour showed a good level of unidirectionality. A significant linear dominance hierarchy, emerged based on the direction of both submissive behaviours ($N = 336$

interactions) and dominance behaviour ($N = 469$ interactions). However, the linearity for both behavioural categories was not high, although submissive and dominance behaviours were both highly unidirectional. Based on these results we applied the I&SI method to both behavioural categories to find the best rank order to fit the linear model. Rank order based on submissive behaviour was positively correlated with the rank order based on dominance behaviour (rs $= 0.85$, $n = 19$, $p = 0.0001$) but the former resulted in an inconsistency of a lower strength than the latter. Finally, both dominance and submissive behaviours showed a high value of triangle transitivity, indicating that the low level of linearity, was due to the high percentage of null dyads (52.05% both) than to a real absence of transitivity.

### Agonistic behaviour in the presence of food

Aggressive interactions ($N = 277$) recorded in the presence of food showed a significant level of linearity but a very low unidirectionality. Both dominance ($N = 272$ interactions) and submissive behaviour ($N = 214$ interactions) also showed a significant but low level of linearity and the highest values of unidirectionality. The rank orders based on aggressive behaviour was correlated to those based on dominance and submissive behaviours (rs $= 0.72$, $n = 19$, $p = 0.0005$; rs $= 0.68$, $n = 19$, $p = 0.001$, respectively) but it resulted in a high number of inconsistencies. The dominance rank order was correlated to the submissive rank order (rs $= 0.75$, $n = 19$, $p = 0.0002$) and both generated one inconsistency with a strength of 2 for submissive behaviours and a strength of 4 for dominance behaviours. All the behavioural categories showed a high triangle transitivity, with dominance behaviours showing the highest value, indicating that, as in the absence of food, the low level of linearity was due to the high percentage of null dyads (aggressive behaviour: 51.86%, submissive behaviour: 60.82%, dominance behaviour: 63.13%) rather than to a real absence of transitivity.

In sum, both dominance and submissive behaviours showed a significant linearity higher values of directionality and triangle transitivity, and a lower number of inconsistencies than aggressive behaviours, resulting in the better measures of dominance relationships.

## Comparison of agonistic dominance in the absence, and in the presence of food

The I&SI rank orders based on submissive behaviours in the two contexts were highly correlated (rs $= 0.81$, $n = 19$, $p = 0.00008$). In the same way, the I&SI rank based on dominance behaviours found in the absence of food was highly correlated with the rank based on dominance behaviours found in the presence of food (rs $= 0.71$, $n = 19$, $p = 0.0007$). We could reasonably assert that the slight differences in the rank orders were probably due to the quite high percentage of null dyads observed in each context. Therefore, dominance relationships between wolves do not appear to be affected by the competitive contexts.

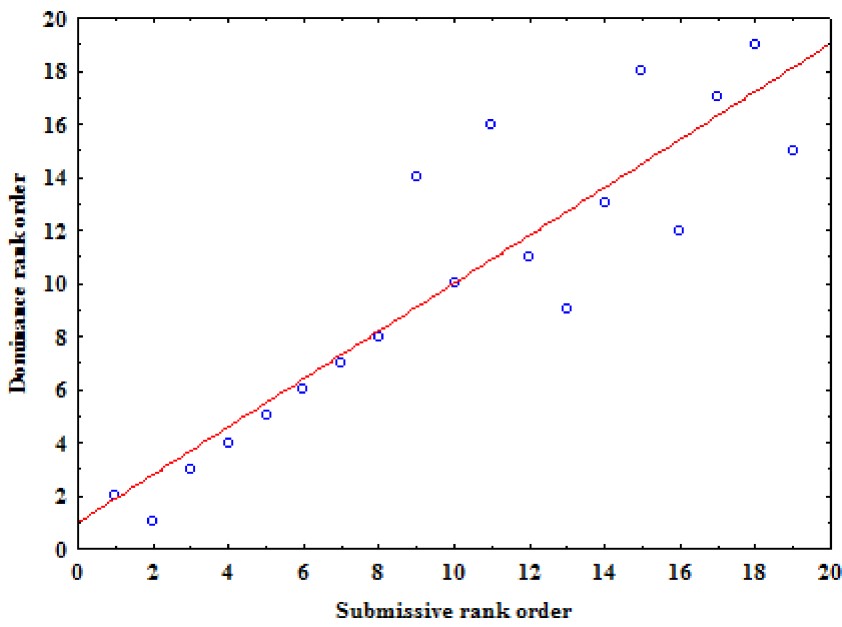

**Figure 1** The relation between the rank order based on all dominance behaviours and the rank order based on all submissive behaviours.

## Which behavioural category is the best indicator of dominance relationships?

Considering that from the previous analyses a high correlation was found in the presence and absence of food and in order to minimize the number of null dyads and hence obtain the most reliable dominance hierarchy, the outcomes of all submissive dyadic interactions ($N = 550$) and all dominance interactions ($N = 741$) were ranked in two different squared matrices. For both submission and dominance interactions we found a high level of directionality, significant levels of linearity and high triangle transitivity (Table 3). By reordering the two matrices following the I&SI method we found two highly correlated rank orders (rs = 0.90, $N = 19, p = 0.0001$; Fig. 1). The alpha male and alpha female were the highest in rank, although they exchanged positions between the two hierarchies, with the alpha female being the highest in rank in hierarchy based on submissive behaviours and the alpha male being the highest in rank in the hierarchy based on dominance behaviours. Nevertheless, the matrix of dominance behaviour generated two inconsistencies with a total strength of 9. Conversely, the matrix of submissive behaviours (Table 4) showed no inconsistencies and revealed the highest triangle transitivity. Therefore, we conclude that submission can be considered the most reliable indicator of linear dominance relationships in our pack.

Excluding the alpha male and female from the analyses, we still found a highly significant linear hierarchy based on both submissive and dominance behaviours (Table 3). Although both linearity and unidirectionality were lower than those obtained including the breeding pair in the analyses, rank orders were precisely the same. The triangle transitivity of dominance behaviours decreased while that of submissive behaviour remained the same, showing again a complete transitivity.

**Table 4  Dominance relationships based on all submissive behavioural patterns recorded between wolves.**

|      | *vik* | mac | mas | sec | sfr | zam | due | **unb** | vol | lac | nor | can | tag | mul | *muc* | *sos* | sto | pro | *hus* |
|------|------|-----|-----|-----|-----|-----|-----|-----|-----|-----|-----|-----|-----|-----|-----|-----|-----|-----|-----|
| **vik** | * | 0 | 0 | 0 | 0 | 0 | 0 | 0 | 0 | 0 | 0 | 0 | 0 | 0 | 0 | 0 | 0 | 0 | 0 |
| **mac** | 1 | * | 0 | 0 | 0 | 0 | 0 | 0 | 0 | 0 | 0 | 0 | 0 | 0 | 0 | 0 | 0 | 0 | 0 |
| **mas** | 1 | 14 | * | 0 | 0 | 0 | 0 | 0 | 0 | 0 | 0 | 0 | 0 | 0 | 0 | 0 | 0 | 0 | 0 |
| **sec** | 3 | 14 | 4 | * | 2 | 0 | 0 | 0 | 0 | 0 | 0 | 0 | 0 | 0 | 0 | 0 | 0 | 0 | 0 |
| **sfr** | 2 | 4 | 13 | 4 | * | 0 | 0 | 0 | 0 | 0 | 0 | 0 | 0 | 0 | 0 | 0 | 0 | 0 | 0 |
| *zam* | 0 | 3 | 0 | 0 | 0 | * | 0 | 0 | 0 | 0 | 0 | 0 | 0 | 0 | 0 | 0 | 0 | 0 | 0 |
| **due** | 1 | 10 | 3 | 2 | 1 | 0 | * | 0 | 0 | 0 | 0 | 0 | 0 | 0 | 0 | 0 | 0 | 0 | 0 |
| *unb* | 48 | 6 | 1 | 0 | 0 | 0 | 1 | * | 0 | 0 | 0 | 0 | 0 | 0 | 0 | 0 | 0 | 0 | 0 |
| **vol** | 31 | 4 | 1 | 2 | 2 | 0 | 0 | 2 | * | 0 | 0 | 0 | 1 | 0 | 0 | 0 | 0 | 0 | 0 |
| **lac** | 27 | 3 | 1 | 2 | 1 | 0 | 0 | 1 | 9 | * | 0 | 0 | 1 | 0 | 0 | 0 | 0 | 0 | 0 |
| **nor** | 3 | 1 | 0 | 0 | 0 | 0 | 0 | 0 | 4 | 6 | * | 0 | 0 | 0 | 0 | 0 | 0 | 0 | 0 |
| **can** | 1 | 2 | 0 | 0 | 0 | 0 | 0 | 3 | 0 | 6 | 0 | * | 0 | 0 | 0 | 1 | 0 | 0 | 0 |
| **tag** | 2 | 13 | 3 | 0 | 7 | 1 | 1 | 0 | 0 | 0 | 0 | 0 | * | 0 | 0 | 0 | 1 | 0 | 0 |
| **mul** | 5 | 22 | 10 | 0 | 10 | 0 | 0 | 1 | 1 | 2 | 0 | 0 | 0 | * | 0 | 0 | 0 | 0 | 0 |
| *muc* | 11 | 13 | 0 | 2 | 2 | 0 | 1 | 1 | 8 | 16 | 1 | 42 | 1 | 1 | * | 2 | 0 | 0 | 0 |
| *sos* | 1 | 2 | 0 | 0 | 1 | 1 | 1 | 0 | 0 | 0 | 0 | 1 | 0 | 0 | 11 | * | 0 | 0 | 0 |
| **sto** | 4 | 4 | 3 | 0 | 1 | 0 | 0 | 1 | 2 | 1 | 2 | 0 | 4 | 0 | 2 | 0 | * | 0 | 0 |
| **pro** | 1 | 23 | 3 | 2 | 5 | 2 | 7 | 1 | 0 | 1 | 0 | 7 | 2 | 5 | 3 | 1 | 1 | * | 0 |
| *hus* | 2 | 1 | 4 | 0 | 0 | 1 | 0 | 0 | 0 | 1 | 0 | 0 | 0 | 1 | 0 | 0 | 0 | 1 | * |

**Notes.**
Bold type, males; italic type, females.
The signallers, who are the performers of the submissive behaviours, are listed in rows, whereas the recipients in columns.

## Can greeting behaviour be considered a signal of formal submission?

Greeting interactions ($N = 182$) showed a significant but very low linearity index (ILT: $h' = 0.34$, $p = 0.008$). Alpha male and the alpha females displayed greetings towards each other. This determined an incomplete, yet still very high, level of unidirectionality (DCI = 0.98). The complete triangle transitivity ($t_{tri} = 1.000$, $p = 0.000$) indicated that the low level of linearity was likely due to high percentage of null dyads (66.08%) and not due to the absence of transitivity. Although the linearity was statistically significant, due to the high percentage of null dyads, the I&SI rank order based on greeting was considered to be unreliable. The individual frequency of greetings displayed was not correlated to the I&SI agonistic rank ($rs = -0.13$, $n = 19$, $p = 0.58$) but the latter was highly and negatively correlated with the frequency of greetings received ($rs = 0.87$, $n = 19$, $p = 0.0001$). In other words, the higher the wolves were in rank, the more greetings they received. In particular, 60.99% of greetings were received by the breeding pair. At the dyadic level, greeting was directed mainly from subordinates towards dominant individuals (GLM: $z = -7.51$, $p < 0.0001$); it also occurred most often between partners with higher age differences (GLM: $z = 5.91$, $p < 0.0001$), with younger wolves displaying the behaviour more often towards the older ones. The sex combination of the dyad did not affect the occurrence of greeting (GLM: $z = 0.37$, $p = 0.71$). Overall, greeting was observed only in relatively few dyads Excluding the breeding pair, in such dyads, greeting

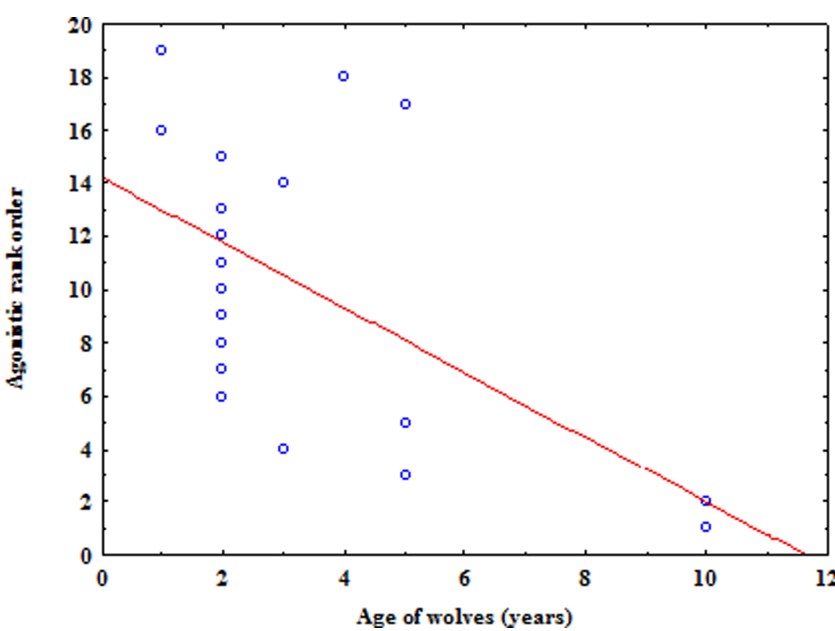

**Figure 2** The relation between the agonistic rank order based on submission and the age of wolves.

was completely unidirectional with subordinate wolves showing this behaviour towards dominant ones.

## Does the rank order in our pack conform to the 'age-(sex)-graded' model?

The agonistic rank order was significantly but only mildly correlated with the age of the wolves (rs $= 0.51, n = 19, p = 0.02$; Fig. 2) while sex had no significant effect on dominance rank ($U = 34, Z = -0.90, p = 0.37$). These results could mainly be due to some adult males who ranked at the bottom of the hierarchy. In particular, two of them (Storto and Procione, 17th and 18th rank position, respectively) showed lower levels of interactions (mean $\pm$ sd of interaction: $3.94 \pm 0.002$) and spent less time close to other wolves (mean $\pm$ sd of the average dyadic association index, Clutton-Brock, 1982: $0.004 \pm 0.002$) than all other pack members (mean $\pm$ sd of interaction: $8.39 \pm 3.60$; mean $\pm$ sd of the average dyadic association index: $0.03 \pm 0.01$). Therefore, they appeared to be peripheral individuals in the pack who probably would have dispersed in the wild. Excluding these two wolves from analyses shows the correlation between rank and age to be noticeably higher (rs $= 0.80, n = 17, p = 0.0001$), while sex still has no effect ($U = 18, Z = -1.73, p = 0.09$). Overall, dominance relationships appeared to be influenced by age, with older wolves being dominant over younger individuals, but not by sex.

However, sex did affect the distribution of agonistic behaviours. The frequency of submissive and aggressive behaviours were higher between female–female (FF) and male–male (MM) dyads than female–male dyads (submissive behaviours, GLM: FF-FM $z = -3.76, p = 0.0004$; MM-FM $z = 4.04, p = 0.0002$; aggressive behaviours, GLM: FF-FM $z = -5.19, p = 0.001$; MM-FM $z = 3.25, p = 0.003$) but no differences emerged between

**Table 5  Different dominance measures.** Summary of values of the directionality (directional consistency index, DCI), linearity h′ and its significance level, number and strength of inconsistencies (No. I and SI, respectively) for the I&SI rank orders, and triangle transitivity (proportion of transitive triangles relative to all trangles $P_t$, triangle transitivity metric $t_{tri}$, and significance level) for all behavioural categories for females and males separately.

| | Directionality | Linearity | Inconsistency: no. I (and SI) | Triangle transitivity |
|---|---|---|---|---|
| **Agonistic behaviours displayed between females** | | | | |
| Female aggressive behaviour | DCI = 0.92 | Yes (h′ = 0.78, p = 0.004) | 0 | $P_t = 1.000$, $t_{tri} = 1.000$, $p = 0.008$ |
| Female dominance behaviour | DCI = 0.98 | Yes (h′ = 0.74, p = 0.007) | 1 (2) | $P_t = 0.973$, $t_{tri} = 0.892$, $p = 0.004$ |
| Female submissive behaviour | DCI = 0.98 | Yes (h′ = 0.59, p = 0.048) | 0 | $P_t = 1.000$, $t_{tri} = 1.000$, $p = 0.008$ |
| **Agonistic behaviours displayed between males** | | | | |
| Male aggressive behaviour | DCI = 0.90 | Yes (h′ = 0.61, p = 0.015) | 1 (4) | $P_t = 0.972$, $t_{tri} = 0.889$, $p = 0.002$ |
| Male dominance behaviour | DCI = 0.96 | Yes (h′ = 0.78, p = 0.0009) | 0 | $P_t = 1.000$, $t_{tri} = 1.000$, $p = 0.001$ |
| Male submissive behaviour | DCI = 0.97 | Yes (h′ = 0.81, p = 0.0002) | 0 | $P_t = 1.000$, $t_{tri} = 1.000$, $p = 0.001$ |

male–male and female–female dyads (submissive behaviours, GLM: FF-MM $z = -0.06$, $p = 0.99$; aggressive behaviours, GLM: FF-MM $z = -1.59$, $p = 0.24$). Female–female dyads showed a frequency of dominance behaviours higher than female–male dyads (GLM: FF-FM $z = -5.20$, $p = 0.001$) but no significant difference was found between all other combinations (GLM: MM-FM $z = 1.96$, $p = 0.11$; FF-MM $z = -2.13$, $p = 0.07$).

Based on these results we analysed the intra-sexual hierarchical organization. The directional consistency (DCI), linearity (h′), the number and strength of inconsistency (I and SI), the proportion of transitive triangles relative to all triangles ($Pt$), and the triangle transitivity metric ($t_{tri}$) for each agonistic behavioural category in the two different contexts analysed (in the presence of food and in its absence) are summarised in Table 5. Aggressive behaviour emerged as the best measure of dominance relationships between females since it showed the highest value of linearity, complete triangle transitivity, and a rank order without any inconsistencies. A similar result was found for submissive behaviours between females, although it showed a low linearity due to the relatively high percentage of null dyads (36.11%). However, the aggressive rank order was highly correlated to the submissive rank order (rs = 0.97, $n = 9$, $p = 0.00002$). The best measure of dominance relationships between males was based on submissive behaviours, which showed the highest values of directionality and linearity, complete triangle transitivity and no inconsistencies in the rank order. In fact, dominance behaviour between males also appeared to be a good measure of hierarchy and the relative rank order was highly correlated to that based on submissive behaviours (rs = 0.94, $n = 10$, $p = 0.00006$).

The female rank orders based on both aggressive and submissive behaviours and the male rank orders based on both submissive and dominance behaviours were highly correlated to the rank order that both sexes had in the entire inter-sexual hierarchy based on submissive behaviours (female aggressive behaviour: rs = 0.98, $n = 9$, $p < 0.00001$; female submissive behaviour: rs = 0.98, $n = 9$, $p < 0.00001$; male submissive behaviour: rs = 0.93, $n = 10$, $p = 0.0001$; male dominance behaviour: rs = 0.99, $n = 10$, $p < 0.00001$)

Female rank orders were positively correlated to age (aggressive rank order: rs = 0.76, $n = 9$, $p = 0.02$; submissive rank order: rs = 0.83, n = 9, p = 0.05). For males positive

and significant correlations were found only when the two potentially dispersing adult males were not considered (all males: submissive behaviour, rs $= 0.38$, $n = 10$, $p = 0.28$ and dominance behaviour, rs $= 0.37$, $n = 10$, $p = 0.29$; without the two dispersing males: submissive behaviour, rs $= 0.78$, $n = 10$, $p = 0.002$ and dominance behaviour, rs $= 0.79$, $n = 10$, $p = 0.002$).

Overall, although no sex effect emerged on the hierarchical rank order of the pack, both males and females showed agonistic behaviours preferentially towards other males and females, respectively. Sex-separate hierarchies showed higher linearity than the hierarchy including the whole pack. Male hierarchical relationships appeared to be based on dominance and submissive behaviours, while female hierarchal relationships appeared to be based on aggressive and submissive behaviours.

## DISCUSSION

Using linearity (*De Vries, 1995*), triangle transitivity (*Shizuka & McDonald, 2014*) matrix-ranking procedures (MatMan; (*De Vries, 1998*), and taking into account both the sex and ages of the wolves in our pack, we found: (1) the existence of a clear linear hierarchy unaffected by the competitive context and which remained solid also when the highest ranking individuals (the breeding pair) were removed from analyses, (2) evidence suggesting the use of 'greeting' as a formalised signal of subordination, and (3) partial support for the age-(and sex)-graded model.

### Agonistic dominance relationships in the presence and absence of food

The main result of the current study is that the relationship between family pack members of the Arctic wolves studied were not randomly distributed but rather, showed a high linear, transitive, and significant hierarchy, which remained constant across both the feeding and nonfeeding contexts. Furthermore, we found that the best indicator of dominance, which resulted in the clearest hierarchical relationship between individuals was the direction of submissive behaviours (e.g., crouch, passive and active submission, flee, etc.). The breeding pair was involved in most of the interactions, as previously reported also in other captive and wild packs (e.g., *Van Hooff & Wensing, 1987*; *Mech, 1999*), and although *Shizuka & McDonald (2015)* point out that linearity of a hierarchy may be 'skewed' due to dominant individuals showing more behaviours than the rest, this was not the case in our study. Indeed results showed that a linear and completely transitive hierarchy based on submissive behaviours was still highly significant when the two top-ranking individuals were removed from the analyses. These results indicate that clear dominance relationships exist among all siblings and confirm submissive behaviours as a more reliable indicator of hierarchical relationships in the pack than aggressive and dominance behaviours.

Although the common social structure in wild wolves is usually made up of the reproducing parent pair and their offspring of the last two years, ranging from two to 15 individuals (e.g., *Bloch, 2002*), families composed of several generations of up to 19–26 individuals (e.g., *Landau, 1993*; *White, 2001*; *VonHoldt et al., 2008*; *Smith et al., 2012*),

have been described. Hence our pack of Arctic wolves, composed of five generations and a total of 19 individuals, can be considered representative of a multigenerational family pack of wolves. Therefore, the clear presence of a linear hierarchy in our family pack of wolves goes against recent suggestions that hierarchical relationships may only be adequate to describe atypical pack structures such as disrupted families or forced packs of unrelated individuals (*Mech & Cluff, 2010*) and rather supports the importance of this concept also in describing the relationship between wolves in a multi generational family pack.

A further confirmation of the importance of dominance relationships in wolves comes from our second finding, that such relationships remained constant across competitive and non-competitive contexts. Indeed the dominance and submissive rank orders detected in the presence of food were highly correlated to the respective rank orders detected in the absence of food. The slight differences observed between the two contexts may be explained by the high percentage of null dyads (*Van Hooff & Wensing, 1987*). Indeed, when adding together all submissive and dominant interactions occurring in the two contexts, we found improved values of linearity and unidirectionality for both hierarchies, further indicating that dominance relationships in our family pack were not influenced by the competitive context. In some mammal species, it may be reasonable to predict the existence of asymmetries in fighting abilities and resource value, especially between different age-sex classes, leading to different rank orders in different contexts. Indeed, food is considered a major determinant of the reproductive success of individuals hence, in species where females play the main role in rearing pups food should have a higher value for them than for males, leading to a female over male dominance hierarchy during feeding competition but not in other contexts. This has been found to be the case in both chimpanzees and cats, where females raise their infants largely with no male intervention (e.g., chimpanzees, *Noë, De Waal & Van Hooff, 1980*; domestic cats, *Bonanni et al., 2007*). However, wolf packs rely on cooperation between all pack members in both rearing pups and providing food; this may account for the consistency of dominance relationships and the absence of a diverse effect of sex in the feeding and non-feeding context.

Submissive behaviours best fulfilled the criteria of agonistic dominance indicators since they showed a higher linearity, a complete transitivity, and rank orders with no inconsistencies. The importance of submissive behaviours in establishing and maintaining dominance relationships have been widely highlighted in primates (*Rowell, 1974*; *De Waal & Luttrell, 1985*) but also in wolves (*Schenkel, 1967*; *Mech, 1999*). In our pack, subordinate individuals often determined the outcome of agonistic interactions by lowering themselves when being approached by or when approaching dominant individuals, as described in wild wolf interactions (*Mech, 1999*). Similar results have also been found in other captive family packs (*Van Hooff & Wensing, 1987*; *Romero et al., 2014*).

In sum, to date the results suggest that submissive behaviours play a more relevant role than dominance displays in terms of maintaining dominance relationships between all members of a family pack, although further investigation should assess the importance of

submissive behaviours in promoting friendly relations and pack cohesion in wolf pack, as has been suggested by some authors (e.g., *Mech, 1999*).

## Greeting as formal indicator of submission

Overall our results showed that greeting in Arctic wolves partially fulfilled the criteria of a formal signal of submission, although it occurred only in a limited number of dyads, it was almost completely unidirectional and it was exhibited in line with the agonistic dominance hierarchy, in that it was displayed mainly by subordinate individuals towards dominant ones. The main exception was the breeding pair, in which this behaviour was exchanged equally. This is particularly interesting considering that their relative position in the hierarchy was also not always fixed; the male appearing dominant over the female when calculating the rank based on dominance displays and *vice versa* when basing the rank on submissive behaviours. This might indicate a relaxed dominance relationship between breeding partners, with the females prevailing in some situations and the male in others, as described in other captive family packs (*Van Hooff & Wensing, 1987*) and wild packs (e.g., *Mech, 1999*).

The frequency of greeting behaviours however, did not correlate with the agonistic rank making it an unreliable measure on which to base the ordering of pack members in a consistent linear hierarchy. This was most likely due to the numerous dyads in which no greeting behaviours were observed. Indeed greeting in wolves is widely described to occur upon reunion after a period of separation and before travelling and hunting (e.g., *Mech, 1999*; *Peterson et al., 2002*). This context is limited in captivity, which may explain why, for many dyads, we were unable to observe this behaviour. Nevertheless, with the only exception of the breeding pair, who exchanged greetings exclusively towards each other, in all other dyads the behaviour was completely unidirectional, displayed mainly from subordinates towards the dominant wolves and with all pack members showing this behaviour most often towards the dominant breeding pair. This is consistent with studies in other canids, showing that the use of greeting is a signal of acknowledgment of dominance status (Ethiopian wolves, *Sillero-Zubiri, Gottelli & Macdonald, 1996*; domestic dog, *Cafazzo et al., 2010*), and with theories suggesting that the mouth licking behaviour that occurs during greeting interactions may be derived from food begging behaviour displayed from the offspring towards the breeding pair to elicit regurgitation (*Schenkel, 1967*; *Mech, Wolf & Packard, 1999*).

## The age-(sex)-graded model

In our pack we found an overall dominance hierarchy based on submissive behaviours in which males were not, on average, higher in rank than females, but older wolves were dominant over younger ones (with the exception of two adult males who ranked at the bottom of the hierarchy). In wild wolf packs, it is usually reported that all members submit to the breeding pair, and the breeding female to the breeding male, with no clear dominance displays being observed between offspring (*Mech, 1999*; *Bloch, 2002*). In captive studies, males are mostly described as being dominant over females and older individuals over younger ones (e.g., *Van Hooff & Wensing, 1987*; *Romero et al., 2014*).

However, differently from the current study, previous work both in captivity and in the wild never statistically tested the effect of sex and age. Our results support the effect of age, showing largely that older siblings are dominant over younger ones, but do not support a sex effect of males being dominant over females.

Interestingly however, when looking at the frequency of the three main behavioural categories used to calculate dominance relationships in our pack, we found that female–male agonistic interactions were fewer compared to intra-sexual (female–female and male–male) agonistic interactions. In other words, although an overall hierarchy including animals of both sexes was detected, agonistic displays were not so frequently expressed in interactions between females and males. In fact, females showed agonistic behaviours preferentially towards other females and males towards other males. Taking into account this differential pattern of behaviours, we calculated sex-separate linear hierarchies, which, in both cases, showed stronger linearity than the mixed hierarchy. Moreover, dominance relationships appeared to be expressed making use of different behavioural categories in male's and female's hierarchies. The best hierarchies (in terms of unidirectionlity, linearity, and transitivity) in females were based on aggressive and submissive interactions, whereas in males, hierarchies based on submission and dominance behaviours showed better indices. Taken together, results suggest that although the pack as a whole shows a clear hierarchical organization, the structure of the hierarchy within each sex is even clearer. Furthermore, it appears that females and males may use different ways to communicate their reciprocal rank when interacting with members of their own sex. *Van Hooff & Wensing (1987)* found similar results in a family pack of European wolves, where intra-sexual relationships were characterized by a higher intensity of exchange of agonistic behaviour than inter-sexual relationships.

An unexpected result is that females, but not males, appeared to use aggression to communicate their reciprocal status in interactions with other females. This result disagrees with most of the studies on hierarchies outlining submission as the best measure of dominance relationships (e.g., *Rowell, 1974*; *Bernstein, 1981*; *Hand, 1986*; *Cafazzo et al., 2010*. A potential explanation is that in general, aggressive interactions were frequent between individuals of our study pack, potentially due to the data being collected mostly during the breeding season, which starts in January and lasts approximately until April. During this time the hierarchical structure of the pack likely regulates breeding activity, and aggressive interactions may be used to more forcefully maintain the status among individuals. Further studies are needed to ascertain whether indeed the behaviours used to maintain the hierarchy are different during breeding and non-breeding periods.

The greater linearity of the hierarchical organization and the differential patterns of behaviours used to maintain it in males and females raises the question about the most appropriate way to characterize the dominance relationships among members of a wolf pack. Are sex-separate hierarchies a better model than all-member hierarchies to describe such relationships? Several authors suggest that separate same-sex hierarchies best describe the social structure of wolf packs (*Schenkel, 1947*; *Rabb, Woolpy & Ginsburg, 1967*; *Zimen, 1975*; *Zimen, 1978*; *Derix et al., 1993*; *Derix & Vanhooff, 1995*). Nevertheless Zimen highlighted the existence of an overall hierarchy with males being dominant over

females in each age class, which is also the usual model reported in studies of wild wolves (*Clark, 1971*; *Mech, 1999*). Unfortunately, as stated above, most of the previous studies carried out both in captivity and in the wild, did not follow a systematic procedure aimed to statistically show the dominance relationships in the pack, which makes a comparative assessment of results difficult. Based on current results, when considering the pack as a whole the hierarchical structure does not show males being dominant over females, but given the stronger linearity indices of separate male and female hierarchies, it would appear that status within sexes may carry an even greater weight than within the mixed sex group.

A final point to consider is the validity of captive-based studies when attempting to characterize the social structure of a wild species. It is undeniable that studies with wild animals are preferable when exploring such topics; however, it is perhaps interesting to note that in a metanalyses involving 113 studies looking at dominance structures in 85 species (172 groups), *Shizuka & McDonald (2015)* found that whether studies were conducted in the wild or in a captive setting did not affect results. With such elusive species as wolves, partial reliance on captive studies is probably unavoidable, however future research using the same methodologies adopted here on wild animals would be particularly important to further our understanding of wolves' social behaviour.

## CONCLUSION

In conclusion, we emphasize the importance of applying a systematic methodology including both age and sex in order to analyse dominance relationships between pack members. Results clearly show that both within each sex and for the pack as a whole, dominance relationships are a meaningful concept, which can be used to describe the structure of a multi-generational family pack of captive wolves. Future studies analysing the potential effects of dominance relationships on other aspects of the animal's lives will likely help to further establish the importance of this concept to describe the social lives of wolves.

## ACKNOWLEDGEMENTS

We are grateful to Dr. Jitka Vokurková, the director and employees of Olomouc Zoo for allowing us to run observations in their institution. Special thanks are due to Daizaburo Shizuka for statistical advice and to Rachel Dale for language revision. We also thank Lesley Rogers and two anonymous referees for useful and constructive suggestions that improved the manuscript.

### Funding

This study was supported by funding from the Austrian Science Fund (FWF): project number M1400-B19 (to Simona Cafazzo) and by the European Research Council under the European Union's Seventh Framework Program (https://ec.europa.eu/research/fp7/

index_en.cfm) by ERC Grant Agreement n. [311870], (to Sarah Marshall-Pescini and Simona Cafazzo). The funders had no role in study design, data collection and analysis, decision to publish, or preparation of the manuscript.

## Grant Disclosures
The following grant information was disclosed by the authors:
Austrian Science Fund: M1400-B19.
European Research Council under the European Union's Seventh Framework Program: 311870.

## Competing Interests
The authors declare there are no competing interests.

## Author Contributions
- Simona Cafazzo conceived and designed the experiments, analyzed the data, wrote the paper, prepared figures and/or tables.
- Martina Lazzaroni conceived and designed the experiments, performed the experiments, contributed reagents/materials/analysis tools, reviewed drafts of the paper.
- Sarah Marshall-Pescini conceived and designed the experiments, contributed reagents/-materials/analysis tools, reviewed drafts of the paper.

## Animal Ethics
The following information was supplied relating to ethical approvals (i.e., approving body and any reference numbers):

The study was purely observational with no manipulation of animals. The relevant committee, Tierversuchs-kommission am Bundesministerium für Wissenschaft und Forschung (Austria) allowed us to run this research without special permissions regarding animals (wolves) since this is not required in such socio-cognitive studies in Austria (Tierversuchsgesetz 2012–TVG 2012).

## Data Availability
The raw data has been supplied as a Supplemental File.

## Supplemental Information
Supplemental information for this article can be found online at http://dx.doi.org/10.7717/peerj.2707#supplemental-information.

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
