# Peer review of "Dominance relationships in a family pack of captive arctic wolves (Canis lupus arctos): the influence of competition for food, age and sex"

_PeerJ, doi:10.7717/peerj.2707_

## Round 0.1 · original submission · Major Revisions

· Academic Editor

Major Revisions

In light of your Appeal, we are willing to consider a revised manuscript in which you address the comments, engage fully with Shizuku and MacDonald (2015) and make all of the other corrections. Of course, it is still possible that the final decision might be rejection.

Hence the submission is returned to you, for you to submit a rebuttal and revision.

· Appeal

Appeal


· · Academic Editor

Reject

I regret to inform you that your manuscript is not suitable for publication. One reviewer has recommended rejection and, although the other reviewer has said only minor corrections, in fact the changes suggested are major.

It is important that you address the recent paper mentioned to you by reviewer 2 and say what is new about your own research.

If you revise your paper taking all of the reviewers' comments into consideration, you may resubmit and begin the reviewing process again.

Reviewer 1 ·

Basic reporting

The English could be improved, but is basically understandable.
The paper in general does not contain major flaws, the introduction and background are sufficient and the structure conforms with PeerJ discipline norm. As the method section is rather clear, the results section needs some clarification (see validity of findings).
There was an error during raw data uploading: sociomatrices and data for GLM were exchanged.

Experimental design

Introduction (line 55):
Please better define “triangle transitivity” just as it was done for other definitions (e.g. “linear hierarchy model”).

Materials & Methods (Line 153): The pack was composed of 20 individuals: 10 males and 10 females, 9 adults (defined as older than two years) and 11 sub-adults (younger than 2 years).
Given that “sex” and “age” are essentials for the manuscript it would be recommended to specify how many males/females were adults/sub-adults.

Validity of the findings

Results:
Line 247: “Rank order based on submissive behaviour was positively correlated with the rank order based on dominance behaviour (rs = 0.85, n = 19, p = 0.0001)”.
Line 257:” The rank orders based on aggressive behaviour was correlated to those based on dominance and submissive behaviours (rs = 0.72, n = 19, p =0.0005; rs = 0.68, n = 19, p = 0.001, respectively) and so on…

The correlations are interesting but please present some plots of the data in cases of significant correlations in order to reassure the reader that the data were not clustered. Clustered data are not suitable for correlation.

From the paper it is meant that subjects were not desexed, is this correct? If this was the case, how was the estrous cycle detected in females? This aspect is fundamental given the well-attested role of estrogens on aggressive behaviors.

Reviewer 2 ·

Basic reporting

The paper conforms in all respects to the rules of basic reporting set out by PeerJ. And the topic concerning dominance hierarchies is also appropriate for the journal since the ubiquity of dominance hierarchies across all animal systems is well established. Although rather wordy, the paper is fine. Its experimental design is ok, although estrous cycles in females change behaviour and no mention is made of this. Some stats may also need to be revisited e.g. –I don’t think clustered data are suitable for correlations?

Experimental design

see above

Validity of the findings

see below

Additional comments

However, I am not convinced that this is a publishable paper. There was an excellent paper published in 2015 in Interface by the Royal Society that reviewed dominance hierarchies (Shizuka D, McDonald DB. 2015 The network motif architecture of dominance hierarchies. J. R. Soc. Interface12: 20150080. http://dx.doi.org/10.1098/ rsif.2015.0080).
It had a number of important observations to make with which the authors of the paper under review here should have engaged. Although earlier publications of the authors of the Interface paper are cited, the important one of 2015 is strangely enough not even mentioned ,even though highly relevant.

Moreover, on the point of hierarchies there are some misgivings as well. I always feel uncomfortable when captive populations are studied for social behaviour because captivity and the forced cohabitation of animals may change behaviour entirely, including structures of dominance relations when formed and maintained in captivity. The way of measuring dominance on offer in this paper may indeed be very useful but this could either be done as a short letter or, if such full debate seems warranted, it would have to take account of the Shizuka/McDonald paper and somehow develop this further. All in all, I would not recommend publication at this time.

---

## Round 0.2 · Minor Revisions

· Academic Editor

Minor Revisions

Thank you for correcting your manuscript to take into account the findings of Shizuke and McDonald (2015) and following their suggestions in your data analysis. This improves the paper.

First I have a couple of minor corrections, as follows:

245-246: Please say what analyses were used to ascertain no clustering and state the statistical values.
548; Correct ‘last’ to ‘lasts’.

Secondly, I sought the comments of one of your reviewers and received the following:

I believe the authors have made a convincing argument that their findings are publishable now they have integrated the paper by Shizuku and McDonald (2015) and have engaged with it.

I have no hesitation now to recommend publication.

Perhaps line 437 can be corrected for the grammar error that spills over into meaning (use of comparative without a comparison—: " is more reliable" (than what?); alternative is to changed the sentence.

Please make these final corrections to your manuscript.

---

## Round 0.3 · accepted · Accept

· Academic Editor

Accept

Please note that, in line 234, it should be 'data were', not 'data was'. This correction could be made in the proofs.